# Cross-sectional study of nutritional intake among patients undergoing tuberculosis treatment along the Myanmar–Thailand border

Karim Damji,[1] Ahmar H Hashmi ![ORCID],[2] Lin Lin Kyi,[3] Michele Vincenti-Delmas,[3] Win Pa Pa Htun,[3] Htet Ko Ko Aung,[3] Tobias Brummaier,[3,4,5,6] Chaisiri Angkurawaranon ![ORCID],[2] Verena I Carrara,[3,4,7] Francois Nosten ![ORCID] [3,4]

For numbered affiliations see end of article.

**Correspondence to**
Dr Chaisiri Angkurawaranon;
chaisiri.a@cmu.ac.th

## ABSTRACT

**Objective** This study summarises nutritional intake among patients with tuberculosis (TB) along the Myanmar–Thailand border according to the local diet.

**Setting** TB clinic along the Myanmar–Thailand border.

**Participants** Cross-sectional surveys of 24-hour food recall were conducted with participants receiving anti-TB treatment. Participants were purposively selected to reflect proportion of age, sex and HIV co-infection based on historical patient records. Out of a total of 28 participants, 20 (71.4%) were men and 5 (17.9%) were co-infected with HIV.

**Primary and secondary outcome measures** The primary outcome compared actual recorded intake to recommended intake. Secondary outcomes compared weight gain and body mass index (BMI) from diagnosis to time of survey.

**Results** There were no significant differences in macronutrient or micronutrient intake by sex or for patients supplementing their rations. Mean treatment length at time of survey was 20.7 weeks (95% CI: 16.5 to 24.8). A significantly higher proportion of women (8/8, 100%) met caloric requirements compared with men (9/20, 45.0%, p=0.010), but few participants met other macronutrient or micronutrient requirements, with no significant differences by sex or for patients supplementing their rations. From diagnosis to the time of the survey, participants averaged significant weight gain of 6.48 kg (95% CI: 3.87 to 9.10) and increased BMI of 2.47 kg/m$^2$ (95% CI: 1.45 to 3.49; p=0.0001 for both). However, 50% (14/28) still had mild or more severe forms of malnutrition.

**Conclusions** This cross-sectional survey of nutritional intake in patients undergoing TB treatment in a sanatorium setting demonstrates the difficulty in sufficiently meeting nutritional demands, even when providing nutritional support.

## INTRODUCTION

The interaction between malnutrition and tuberculosis (TB) has long been established.[1] Not only does malnutrition predispose individuals to TB infection, poor nutritional status and weight loss—particularly loss of

### Strengths and limitations of this study

► Strong methodology tailored to the local diet to accurately assess nutritional intake.
► Limited in sample size and not a controlled study of a nutritional intervention.
► Costlier measures that more precisely estimate nutritional intake were not possible in this study.

lean body mass—are also associated with TB.[2–4] Malnutrition at the time of TB diagnosis increases mortality risk.[2 5 6] Inadequate nutritional intake is the most likely cause of poor nutritional status among patients with TB.[1] However, there is insufficient evidence to determine whether nutritional supplementation improves outcomes in patients with TB.[3 7 8] Even in situations where anti-TB therapy and nutritional support is offered, many patients do not return to adequate nutritional status by the end of their treatment period.[1 9] However, in light of the strong link between malnutrition and TB, the WHO insists on nutritional assessment, counselling and addressing nutritional deficits for patients with TB.[3 5 8 10 11]

However, it is difficult to assess the nutritional intake of patients with TB to detect underlying nutritional deficits. Nutritional assessments commonly use 24-hour food recalls or food frequency questionnaires. These tools rely on food composition data to estimate nutrient intake. In low-income and middle-income countries—where most of the global TB cases occur—the lack of nutrient composition data based on local diets leads to inaccurate estimates of nutrient intake.[11] Intake assessments that do not account for local diets and rely on patient recall may not accurately depict nutritional intake of

patients with TB in outpatient or community directly observed therapy (DOTS) programmes.[2 10]

Along the Myanmar–Thailand border, TB clinics provide services for migrants with TB via a unique, sanatorium model.[12–14] Nutritional support in the way of rations are included in the cadre of care and treatment services provided to patients who reside at the clinics for the duration of their treatment. Recent nutrition research has allowed a better characterisation of the local diets along the border sufficient to determine nutritional content.[15 16] This cross-sectional study seeks to quantify macronutrient and micronutrient intake for a sample of patients with TB along the border and to evaluate if meal rations adequately address patients' nutritional demands.

## METHODS

### Setting: TB programme along the border

Established in 2010, the Borderlands Health Foundation/ Shoklo Malaria Research Unit (SMRU) TB programme has provided a large breadth of services for patients with TB through its clinics, including diagnostic and treatment procedures, housing, dietary and psychosocial support services. By the end of January 2019, these clinics had provided treatment services to a total of 1925 patients, 1465 (76%) of whom had successfully completed the entirety of their treatment. At the time of this study, 118 patients were actively receiving treatment in TB clinics along the border. Patients are agricultural migrants from either side of the Myanmar–Thailand border and are predominantly of Burman or Karen ethnicity.

### Study participants and design

To provide a representative sample of patients to enrol in this study, TB programme patient records were reviewed to estimate the proportion of men and women, age and those co-infected with HIV among the patient population. These proportions were then applied to the existing roster of patients currently receiving care at the time of this study. Thus, participants were selected through purposive sampling from participating TB clinics based on proportions representative of patient profiles from historical TB programme records. Inclusion criteria were: TB confirmed by GeneXpert and/or AFB smear, with pulmonary or extrapulmonary infections, multidrug-resistant TB or HIV co-infection. Participants were excluded if they were in treatment for less than 8 weeks, had a recent acute illness onset, unable to provide accurate 24-hour food recalls or unable to provide consent/ verbal assent. Participants meeting inclusion/exclusion criteria were approached and provided verbal assent for this study, as informed consent had been waived for this audit of services in a vulnerable population where legal status needs to be kept confidential. Routine treatment was provided to all patients according to WHO guidelines. As this was not a controlled study testing an intervention but assessing nutritional intake in a small subset of participants, no sample size calculation was performed.

### Nutrition support provided

Patients at TB clinics are provided daily meals for the duration of their treatment. Rations provided by the clinic include two full meals a day as well as refreshments (processed foods and liquids). Some participants were able to supplement their diet with food from outside sources in addition to the rations provided. In addition to daily meals, patients receive weekly rations which include biscuits, juice and sweetened milk powders. Each meal consisted of a serving of rice and a curry. Protein sources for meals come from animal sources every day out of the week, with a predominance of chicken and fish. The meals were provided in timing and frequency according to local eating habits. While meal ingredients are budgeted for each individual patient, servings are not standardised. Chefs were locals from the communities and prepared meals according to local dietary customs and preferences.

### Survey tools and validation

The 24-hour food recalls determined participant food intake and were adapted from a form validated for use in an observational cohort, 'Molecular signature of Karen and Burmese pregnant women on the Thailand-Myanmar border', conducted at SMRU (ClinicalTrials. gov Identifier: NCT02797327).[15] However, as portion size estimations—using visual and other size estimation aids or household utensil measures—are limited,[17–19] caloric intake is often underestimated, such as for cereal grains and other food items.[19] To account for this potential limitation, dietitians advocate the use of multiple methods to estimate portion sizes in 24-hour food recalls such as hand measurements, which have been validated for accuracy and appropriateness of this method for use in low-resource settings.[15] In previous 24-hour food recalls in migrant clinics along the Myanmar–Thailand border, patients often underestimated their intake of rice— similar to studies in Africa.[19] Therefore, 24-hour food recalls have employed all methods at our disposal: standardised measuring instruments, hand measurements and visual aids.[17] Using multiple methods of portion size estimation may enhance the accuracy of recall data.

### Calculating nutritional intake

Following survey completion, food recall data was uploaded into a free, online nutrition data software (Cronometer, https://cronometer.com/). Nutrition data was calculated for the total intake, intake from rations alone and intake excluding rice.

Recipes for all meals provided at the clinic as part of the rations were compiled by interviewing the cooks at the clinic and corroborating common recipes with several local staff at SMRU. This allowed compiling basic recipes for curries, soups, salads, stir fries and pastes. Hence, any dish or meal reported as part of the rations allowed for the main ingredients to be determined and used in conjunction with the compiled recipes. These final recipes were then added to Cronometer.

As several ingredients in this community's diet were not available on Cronometer's database, nutrient data were obtained using the Mahidol University's Thai Food Composition Database and McGill University's Center for Indigenous Peoples' Nutrition and Environment Karen food composition database.[20 21] Nutritional information in these databases is often expressed in terms of nutrients per 100 g sample and required recording densities of ingredients reported in recalls. This was determined either through direct measurement, matching densities to foods with similar descriptions or averaging the densities of similar ingredients.

## Anthropometric measurements and determining nutritional requirements

Anthropometric measurements were performed prior to beginning anti-TB treatment and repeated for each participant immediately prior to conducting the 24-hour food recall and followed TB programme standard of procedures for weight and height assessment. Routinely calibrated mechanical scales were used to measure weight and height accurate to 500 g and 5 mm, respectively. Body mass index (BMI) was calculated and categorised into: severe malnutrition ($<16.0\,kg/m^2$), moderate malnutrition ($16.0$–$16.9\,kg/m^2$), mild malnutrition ($17.0$–$18.4\,kg/m^2$), normal ($18.5$–$24.9\,kg/m^2$) and overweight ($25.0$–$29.9\,kg/m^2$).[22]

Nutritional requirements were calculated for each individual patient and compared to the participants' actual intake. For nutritional requirements of patients with TB with no comorbidities, a protein range of 1.2–1.5 g/kg ideal body weight (IBW) was used. The energy range was 35–40 kcal/kg body weight. IBW was calculated using the Devine Equation (50 kg + 2.3 kg × (height − 60 inches) for men, and 45.5 kg + 2.3 kg × (height − 60 inches) for women), which was adapted for the metric system.[23] For patients with an HIV co-infection, the 35–40 kcal/kg range was increased by 20%–30%. The estimated energy needs of the one participant with an above-the-knee amputation were reduced by 16%. National Academy of Medicine recommendations were used for carbohydrates and fats[24] and the Thai daily recommended intake for micronutrients (vitamins A, B complex, C, calcium, iron, sodium and zinc).[25]

## Statistical analysis

Analysis was conducted using Stata V.14 (StataCorp LP). Distributions of the data were assessed for normality, with descriptive data summarised using mean and 95% CIs or median and IQR as appropriate. The primary outcome of this study was to evaluate participants' dietary intake and their individual nutritional requirements. The secondary outcome assessed weight gain and change in BMI from the start of anti-TB treatment until the time of the survey.

Weight gain, change in BMI and differences in participant intake for all three macronutrients (protein, carbohydrates and fats) and selected micronutrients were evaluated by gender and type of ration using Student's t-test or Mann-Whitney U test for normally distributed and non-parametrically data, respectively. Intake for each nutrient was then categorised as 'dietary requirement achieved' or 'not achieved' and compared by gender and type of ration with Fisher's exact test. The limited sample size prevented performing meaningful multivariable analysis.

## Patient and public involvement in research

Prior to conducting this study, the research team involved the local population through a local community advisory board, the Tak Community Advisory Board, Thailand. Comprised of community leaders, they advised on study design and implementation and approved the study (TCAB 201903). No study participants were directly involved in the design of the study. Although we have not disseminated findings to the study participants who have completed treatment and returned to their communities by the end of the study, study findings have been disseminated to staff to inform meal rations provided in the TB clinics and to inform nutrition counselling for future patients with TB receiving care at the clinic.

## RESULTS

Table 1 summarises participants' basic demographics, treatment and anthropometric variables for all 28 participants. More men were included in the analysis (20/28, 71.4% vs 8/28, 28.6%). Additionally, the men were significantly older than women (52.4 vs 37.6 years, p=0.021). A total of 17.9% (5/28) of participants were co-infected with HIV. None of the participants reported substance use such as alcohol, cigarettes or other drugs. Just over half of the participants (57.1%, 16/28) ate only the rations provided by the clinic. No participants had multidrug-resistant TB, and only one male participant (1/20, 5%) had an extrapulmonary form of TB. At diagnosis, men were significantly taller (160.7 vs 150.8 cm, p=0.002) and heavier (43.2 vs 36.9 kg, p=0.011) than women, but there were no significant differences in BMI (16.7 vs $16.3\,kg/m^2$, p=0.630). All participants were currently receiving treatment with a mean treatment length at the time of survey of 20.7 weeks (95% CI: 16.5 to 24.8). Pairwise comparisons show 22/28 (78.6%) participants had increased weight and BMI from diagnosis to the time of the survey; with an average gain of 6.48 kg (95% CI: 3.87 to 9.10) and $2.47\,kg/m^2$ (95% CI: 1.45 to 3.49) per participant (p=0.0001 for both). By the time of the survey, the weight of men and women were no longer significantly different (51.0 vs 44.0 kg, p=0.115). This improvement in weight and BMI is reflected in the shift in malnutrition categorisation from diagnosis to the time of survey: at the time of diagnosis 89% (25/28) of participants had mild or more severe forms of malnutrition compared with 50% (14/28) of participants at the time of the survey (p=0.22).

Table 2 outlines the average nutritional intake stratified by sex and whether participants ate rations alone. There were no significant differences in total macronutrient or

**Table 1** Basic demographic, treatment and anthropometric variables

| Variables | All participants (N=28) | Male* (20, 71.4%) | Female (8, 28.6%) | |
|---|---|---|---|---|
| Age (mean, 95% CI, years) | 48.2 (42.1 to 54.2) | 52.4 (45.9 to 58.9) | 37.6 (24.8 to 50.5) | p=0.021 |
| Ration only (n, %) | | | | p=0.69 |
| Yes | 16 (57.1) | 12 (60.0) | 4 (50.0) | |
| No | 12 (42.9) | 8 (40.0) | 4 (50.0) | |
| HIV co-infection (n, %) | | | | p=1.00 |
| Positive | 5 (17.9) | 4 (20.0) | 1 (12.5) | |
| Negative | 23 (82.1) | 16 (80.0) | 7 (87.2) | |
| Treatment length at time of survey (mean, 95% CI, weeks) | 20.7 (16.5 to 24.8) | 20.1 (15.4 to 24.8) | 22.2 (11.4 to 32.9) | p=0.628 |
| Height (mean, 95% CI, cm) | 157.8 (154.6 to 161.1) | 160.7 (157.4 to 163.9) | 150.8 (144.4 to 157.2) | p=0.003 |
| Weight at diagnosis (mean, 95% CI, kg) | 41.4 (39.1 to 43.8) | 43.2 (40.7 to 45.8) | 36.9 (31.9 to 41.8) | p=0.011 |
| Weight at survey (mean, 95% CI, kg) | 49.0 (44.9 to 53.0) | 51.0 (46.3 to 55.6) | 44.0 (35.1 to 52.9) | p=0.115 |
| BMI at diagnosis (mean, 95% CI, kg/m$^2$) | 16.1 (15.7 to 17.5) | 16.7 (15.7 to 17.8) | 16.3 (13.8 to 18.7) | p=0.630 |
| BMI at survey (mean, 95% CI, kg/m$^2$) | 19.5 (18.3 to 20.7) | 19.7 (18.3 to 21.1) | 19.1 (16.3 to 21.9) | p=0.682 |

*Includes one participant with an amputation of one leg and one with extrapulmonary tuberculosis.
BMI, body mass index.

micronutrient intake when comparing men to women, or patients eating only the rations provided compared with those supplementing the rations received.

Table 3 highlights the proportion of participants meeting their recommended daily allowances. Participants' diets were low in protein, fats and nearly all vitamins, but were high in salt content. A significantly higher proportion of women met their overall caloric and vitamin C requirements (p=0.010 and p=0.044, respectively), while a higher proportion of men met their iron intake requirements (p=0.011). Meeting the remaining dietary requirements did not differ significantly between the sexes or having a ration-only diet.

## DISCUSSION

This study presents findings from a cross-sectional survey of patients with TB undergoing treatment while residing at a sanatorium-style clinic along the Myanmar–Thailand border. The most notable finding in this study is the difficulty patients had reaching their nutrition requirements. Patients with TB supplementing the rations provided by the clinic were no more likely to meet their requirements than those eating only the provided rations. Frequently, total calories and protein requirements were not met, and although carbohydrates approached the needed requirements, most of the total carbohydrates were acquired from rice, reflecting a lack of dietary diversity. This lack of dietary diversity is likely responsible for participants' inability to meet most micronutrient requirements. Similar to other study findings, although there was some weight gain, improvement in BMI and malnutrition status, this fell below the level desired.[1 8 9] Overall intake of almost all important micronutrients was limited—with sodium intake often being too high—even for patients supplementing the rations provided, further suggesting that participant diets lack the diversity required to meet nutritional recommendations.

These inadequacies in nutrient intake among patients with TB echoes findings in other studies.[1 2 7 9 26] This study adds to the literature by including robust data on the nutritional content of local diets; provides strong nutritional intake data given the sanatorium-style setting; and provides baseline data for a vulnerable, migrant population that likely has worse nutritional status upon diagnosis than may be seen in other populations. Unlike other settings where nutrient intake was estimated in accordance to the local diet, patients with TB in Georgia were able to manage high caloric intake for follow-up in an outpatient setting.[10]

In this setting in Asia, it may be useful to highlight some recommendations for increasing nutritional intake: a regular source of protein; small yet regular meals; and locally sourced fresh fruits and vegetables or other snacks low in sodium and saturated fats that provide additional micronutrients . These recommendations should be considered alongside regular dietary counselling and support—if human resources can be allocated—as recommended by the WHO.[8] Not only are patients recommended to have an assessment when enrolled for care but support may also be important at the beginning stages of drug treatment, when side effects such as nausea, vomiting and loss of appetite can interfere with early nutritional gains.[26] A culturally-tailored counselling programme, representative of the local diet, can help inform patients with TB

**Table 2** Nutrition intake reported as median (IQR)

| | Totals (N=28) | Male (n=20) | Female (n=8) | Ration only (n=16) | Ration+ (n=12) |
|---|---|---|---|---|---|
| Calories (kcal) | 2153.5 (1764.0, 3118.0) | 2065.0 (1487.5, 3118.0) | 2325.0 (1986.5, 3158.5) | 2035.0 (1454.5, 2721.5) | 2153.5 (1968.0, 3638.0) |
| Calories from rice (kcal) | 967.0 (789.0, 1396.5) | 967.0 (842.0, 1450.5) | 967.0 (698.5, 1396.5) | 967.0 (644.5, 1397.0) | 967.5 (789.0, 1396.5) |
| Protein (kcal) | 277.6 (192.6, 371.6) | 268.6 (178.4, 371.6) | 305.8 (225.8, 421.4) | 268.6 (163.6, 360.8) | 302.4 (225.8, 422.0) |
| Carbohydrates (kcal) | 1240.4 (1061.6, 1718.0) | 1248.0 (1025.6, 1891.0) | 1231.8 (1098.0, 1522.0) | 1160.2 (1008.6, 1718.0) | 1313.8 (1173.8, 1791.8) |
| Fats (kcal) | 540.9 (385.7, 797.4) | 455.9 (383.9, 736.7) | 756.9 (526.5, 982.4) | 464.9 (356.4, 711.0) | 736.7 (403.2, 947.7) |
| Vitamin A (mcg RAE) | 293.8 (150.5, 459.4) | 226.8 (150.5, 459.4) | 360.7 (150.6, 494.2) | 226.8 (1510.5, 425.6) | 398.8 (141.9, 502.6) |
| Vitamin $B_1$ (mg) | 1.0 (0.7, 1.5) | 1.0 (0.8, 1.5) | 1.1 (0.7, 1.5) | 1.1 (0.8, 1.3) | 0.9 (0.7, 1.7) |
| Vitamin $B_2$ (mg) | 1.2 (0.7, 1.6) | 1.2 (0.7, 1.6) | 1.2 (0.6, 1.7) | 1.1 (0.7, 1.6) | 1.3 (0.8, 2.1) |
| Vitamin $B_3$ (mg) | 24.2 (17.6, 36.1) | 25.7 (16.9, 38.2) | 23.8 (20.0, 32.5) | 25.9 (16.9, 38.2) | 24.0 (19.4, 32.8) |
| Vitamin $B_5$ (mg) | 3.5 (1.3, 4.4) | 3.0 (1.3, 4.2) | 3.7 (2.1, 5.6) | 3.4 (1.6, 4.2) | 3.7 (0.9, 4.9) |
| Vitamin $B_6$ (mg) | 1.7 (1.0, 2.3) | 1.65 (1.0, 2.3) | 1.7 (0.9, 2.1) | 1.9 (1.0, 2.2) | 1.6 (1.0, 2.5) |
| Vitamin $B_9$ (mcg) | 234.0 (116.5, 303.4) | 205.1 (116.5, 291.9) | 268.9 (142.1, 324.0) | 203.2 (116.5, 293.7) | 280.3 (119.4, 321.5) |
| Vitamin $B_{12}$ (mcg) | 4.3 (1.0, 6.0) | 4.1 (1.0, 5.3) | 5.8 (1.7, 8.0) | 3.5 (1.5, 5.9) | 4.7 (0.8, 6.4) |
| Vitamin C (mg) | 65.2 (48.4, 132.7) | 59.8 (43.1, 111.0) | 112.7 (73.8, 170.8) | 62.2 (47.7, 126.9) | 79.5 (48.4, 132.7) |
| Calcium (mg) | 586.5 (365.8, 906.4) | 619.9 (412.2, 906.4) | 465.5 (292.5, 951.8) | 496.0 (332.5, 690.1) | 642.4 (417.2, 1138.8) |
| Iron (mg) | 14.0 (8.0, 18.4) | 14.7 (8.6, 18.4) | 10.9 (6.5, 18.2) | 11.0 (7.2, 19.4) | 15.5 (10.7, 18.4) |
| Sodium (mg) | 4011.9 (2481.2, 5564.5) | 4011.9 (2283.2, 5359.7) | 4419.4 (3032.8, 5759.0) | 3727.5 (2848.5, 5359.7) | 4429.6 (2230.9, 6245.9) |
| Zinc (mg) | 6.4 (3.1, 10.0) | 5.8 (3.1, 10.0) | 8.4 (3.3, 12.0) | 5.8 (4.2, 9.0) | 8.1 (1.4, 10.7) |

Thai Dietary Reference Intake: vitamin A: 700 mcg RAE males ≥19 years, 600 mcg RAE females ≥19 years; vitamin $B_1$: 1.2 mg males ≥13 years, 1.1 mg females ≥19 years; vitamin $B_2$: 1.3 mg males ≥13 years, 1.1 mg females ≥19 years; vitamin $B_3$: 16 mg males ≥13 years, 14 mg females ≥13 years; vitamin $B_5$: 5 mg males, females ≥13 years; vitamin $B_6$: 1.3 mg males 19–50 years, 1.7 mg males ≥51 years, 1.3 mg females 19–50 years, 1.5 mg females ≥51 years; vitamin $B_{12}$: 2.4 mcg males, females ≥16 years; vitamin C: 100 mg males ≥19 years, 85 mg females ≥19 years; calcium 800 mcg males/females 19–50 years, 1000 mcg males/females ≥51 years; folate: 300 mcg males/females ≥13 years; iron 11.5 mg males 19 60 years, 11 mg males ≥61 years, 20 mg females 19–50 males, 10 mg females ≥51 years; sodium: all participants require less than 2500 mg per day; zinc: 11.6 mg males 19–30 years, 10.9 mg males 31–70 years, 10.3 mg males ≥71 years, 9.7 mg females 19–30 years, 9.2 mg females 31–60; 8.6 mg females ≥61 years.[24]
RAE, retinol activity equivalents.

**Table 3** Percentage (%) of participants meeting their daily requirements

| | Total (N=28) | Male (n=20) | Female (n=8) | Ration only (n=16) | Ration+ (n=12) |
|---|---|---|---|---|---|
| Calories* | 60.7 | 45.0 | 100.0† | 56.3 | 66.7 |
| Protein | 53.6 | 45.0 | 75.0 | 43.8 | 66.7 |
| Carbohydrates | 89.3 | 95.0 | 75.0 | 93.8 | 83.3 |
| Fats | 60.7 | 60.0 | 62.5 | 56.3 | 66.7 |
| Vitamin A | 3.6 | 0.0 | 12.5 | 6.3 | 0.0 |
| Vitamin $B_1$ | 42.9 | 40.0 | 50.0 | 43.8 | 41.7 |
| Vitamin $B_2$ | 53.6 | 55.0 | 50.0 | 50.0 | 58.3 |
| Vitamin $B_3$ | 89.3 | 85.0 | 100.0 | 81.3 | 100.0 |
| Vitamin $B_5$ | 17.9 | 10.0 | 37.5 | 12.5 | 25.0 |
| Vitamin $B_6$ | 57.1 | 55.0 | 62.5 | 56.3 | 58.3 |
| Vitamin $B_9$ | 25.0 | 20.0 | 37.5 | 25.0 | 25.0 |
| Vitamin $B_{12}$ | 64.3 | 60.0 | 75.0 | 62.5 | 66.7 |
| Vitamin C† | 42.9 | 30.0 | 75.0‡ | 37.5 | 50.0 |
| Calcium | 21.4 | 20.0 | 25.0 | 12.5 | 33.3 |
| Iron‡ | 64.3 | 80.0§ | 25.0 | 62.5 | 66.7 |
| Sodium§ | 25.0 | 30.0 | 12.5 | 18.8 | 33.3 |
| Zinc | 21.4 | 20.0 | 25.0 | 18.8 | 25.0 |

*Higher proportion of women met caloric requirements than men (p=0.010, Fisher's exact test).
†Higher proportion of women met vitamin C requirements than men (p=0.044, Fisher's exact test).
‡Higher proportion of men met iron intake requirements than women (p=0.011, Fisher's exact test).
§Proportion of participants with sodium consumption below 2500 mg/24 hours versus those with high sodium intake.

about the importance of a diverse diet while undergoing treatment and where local community awareness about good nutrition may be limited. Including family members in these discussions would be helpful as family members often stay with patients in the clinic. Such counselling can also help inform patients with TB (and their caregivers) of nutritional foods should they be able to purchase foods to supplement their rations.

This study is not without limitations. As this was not a controlled intervention, we did not perform a sample size calculation and instead elected to take a small subset of patients with TB to participate in this study such that it could be conducted feasibly and quickly with the staff on hand. In addition to its small size, this study is limited as it is cross-sectional in design—although this is the design reported by most studies of this nature. With further funding, future studies may assess nutritional intake in addition to blood and serum measures of nutritional status among a larger sample of patients, as the free software we used for this study limited our analysis to only six micronutrients. The comparisons drawn in this sample are limited as well, given the broad inclusion criteria (eg, including participants with HIV co-infection) that were chosen to include a more representative sample of patients from this border region to inform future nutritional interventions. A further limitation in participants assessed in this study is that relapsed patients with TB and those with comorbidities such as diabetes mellitus were not included, as their treatment and nutritional recommendations may differ

from what is commonly prescribed for patients with TB. The significant difference in gender in our study, however, is representative of the greater number of TB infections among men in the general population. Future studies may also benefit from using additional anthropometric measures, especially those more sensitive in detecting undernutrition[6 9 27 28] and that account for disease severity.[25] Some limitation is inherent in the methods used such as not performing serial food recall surveys, measurement intake accuracy and recall bias,[17 18 29] and aligning food descriptions from the local diet with food composition databases. However, the tools used have been developed across different nutritional studies in this area and have allowed for validating methods across different segments of the local migrant populations.

## CONCLUSIONS

This cross-sectional survey of patients with TB residing in a sanatorium setting along the Myanmar–Thailand border demonstrates the difficulty in sufficiently meeting nutritional demands, even when providing nutritional support.

**Author affiliations**
[1]Family and Consumer Sciences, California State University, Northridge, California, USA
[2]Department of Family Medicine, Faculty of Medicine, Chiang Mai University, Chiang Mai, Thailand
[3]Shoklo Malaria Research Unit, Mahidol-Oxford Tropical Medicine Research Unit, Faculty of Tropical Medicine, Mahidol University, Mae Sot, Thailand

⁴Centre for Tropical Medicine and Global Health, Nuffield Department of Medicine, University of Oxford, Oxford, UK
⁵Swiss Tropical and Public Health Institute, Basel, Switzerland
⁶University of Basel, Basel, Switzerland
⁷Institute of Global Health, Faculty of Medicine, University of Geneva, Geneva, Switzerland

**Correction notice** This article has been corrected since it was published. Affiliations of Dr. Tobias Brummaier and Dr. Verena I Carrara have been updated.

**Acknowledgements** We are grateful for the patients who volunteered to be a part of this study. We would like to acknowledge the staff working in the tuberculosis (TB) department at Shoklo Malaria Research Unit (SMRU), for their help and support with logistics, materials and other needs. We would like to thank the local SMRU staff that helped with recipes and understanding local diets, including the cooks at the TB sanatorium. We would like to thank Wolfgang Stuetz for feedback on drafts of this manuscript.

**Contributors** KD, AHH, MV-D, CA, VC, FN: study conception, design. KD, AHH, LLK, WPPH, HKKA, TB: data collection and study implementation. KD, AHH, VC: data analysis/interpretation. All authors critically revised the manuscript and approved the final version submitted and agree to be accountable for this work. AHH and CA are responsible for the overall content as the guarantor.

**Funding** This research was funded in whole, or in part, by the Wellcome Trust Thailand/Laos Major Overseas Program Renewal (grant number 106698). For the purpose of open access, the author has applied a CC BY public copyright licence to any Author Accepted Manuscript version arising from this submission. The funding body had no role in the design of the study and collection, analysis and interpretation of data and in writing the manuscript.

**Competing interests** None declared.

**Patient and public involvement** Patients and/or the public were involved in the design, or conduct, or reporting, or dissemination plans of this research. Refer to the Methods section for further details.

**Patient consent for publication** Not applicable.

**Ethics approval** This study received ethical approval from the Research Ethics Committee of the Faculty of Medicine, Chiang Mai University (FAM-2562-06197) in accordance with the Declaration of Helsinki. Informed verbal assent was given by all participants prior to survey, with written informed consent waived by the Research Ethics Committee of Chiang Mai University.

**Provenance and peer review** Not commissioned; externally peer reviewed.

**Data availability statement** Data are available upon reasonable request. The data sets generated and/or analysed during the current study are not publicly available due to privacy/safety for migrant participants, but are available from the corresponding author on reasonable request.

**ORCID iDs**
Ahmar H Hashmi http://orcid.org/0000-0002-1592-0606
Chaisiri Angkurawaranon http://orcid.org/0000-0003-4206-9164
Francois Nosten http://orcid.org/0000-0002-7951-0745

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
