## [Reviewer comments · BMJ Open]

ARTICLE DETAILS

TITLE (PROVISIONAL)	A cross-sectional study of nutritional intake among patients undergoing tuberculosis treatment along the Myanmar-Thailand border
AUTHORS	Damji, Karim; Hashmi, Ahmar; Kyi, Lin Lin; Vincenti-Delmas, Michele; Htun, Win Pa Pa; Ko Ko Aung, Htet; Brummaier, Tobias; Angkurawaranon, Chaisiri; Carrara, Verena; Nosten, Francois

VERSION 1 – REVIEW

REVIEWER	Zekariyas Sahile Ambo University
REVIEW RETURNED	19-Jun-2021

GENERAL COMMENTS	Title: A cross-sectional study of nutritional intake among patients undergoing tuberculosis treatment along the Myanmar-Thailand border The researchers have selected a good and interesting research title. This type of research is highly needed for the scientific community. However, this research manuscript is not clear methodologically and may not be replicable. The below list of comments may help the researchers to revise the manuscript. Abstract • The study design was not reported in the abstract section.• The researchers need to check the English grammar in the abstract section.• The researcher did not report how the participants were selected, was it purposive sampling.• A total of 28 participants were enrolled in the study and 20 were male and 5 were HIV coinfectd. However, the researcher was mentioned there were 118 participants under TB treatment during this study is conducted. What was the scientific base for small size?• The primary and secondary outcome measure: the reader may not be clear on which outcome is primary and which one is secondary in the Abstract section.• The authors reported they were considered sex, age, and TB/HIV co-infection in participant selection from historical records. What other factors like relapsed TB, MDRTB, and other non-communicable diseases such as diabetes mellitus which is common TB patients. One advantage of stratification is to get an adequate sample from each stratum. However, the researcher report there was a significant difference in the proportion of male and female.• The researchers report there was no significant difference in macro or micronutrient intake by sex or for patients supplementing meal rations. only 4 participants were female. Do you think it appropriate to do a statistical analysis? Did the researchers check the statistical assumptions? Was measure of association the aim for this study?• The researcher also said a statistically significant higher
--

	proportion of women (8/8) met caloric requirements versus men (9/20) and report P-value. Which statistical analysis the researcher performed for this report?  In the abstract section, this statement is not clear "From diagnosis to the time of 43 the survey, although participants had significant weight gain averaging 6.48 kg (95% CI: 44 3.87, 9.10) and 2.47 kg/m² (95% CI: 1.45, 3.49) in BMI (p=0.0001 for both), 50% (14/28) 45 still had mild or more severe forms of malnutrition." The researcher's conclusion is very general in the abstract section. It would be good to focus on which macro or micronutrients were not met the nutritional requirement? It is good to include a very clear description of nutritional support provided for the patients. Because half of the participants had not to access additional sources. I would suggest if the researcher's focus is on the nutritional support and met the nutritional requirement Strength and limitation  The researcher reports they used a strong methodology and accurate assessment of nutritional intake as a strength of the study. In meantime, they report they have not used the more precise measure of nutritional intake as a limitation. Method section  BMI classification in the method section needs a reference. The recommended dietary intake for infected Tuberculosis patients should not be the same as healthy individuals. Did the researcher use the same RDA as healthy individuals? This is not described in the report. The researcher calculated the protein for macronutrient requirement for TB and TB/HIV by sex. The recommended requirement for carbohydrates and fat were not included in the manuscript report. What is the scientific base for selecting 6 micronutrients? Which specific statistical analysis used to find associations? It is not clearly described in the statistical analysis section. Result  The researcher reported that all participants were received two full meals per day and refreshments (processed foods and liquids). Half the participants ate the ration only from the clinic. If this information is known what is the need to ask the patient for a 24-hour recall? I think it would be good if the researchers were assessed the nutritional supplementation was met the nutritional requirement for Tuberculosis patients and its impact. Because, as I understand, most of the patients in that setting are mostly dependent on nutritional support provided by the clinic.
--	--

REVIEWER	Leslie Chavez-Galan Instituto Nacional de Enfermedades Respiratorias, Integrative Immunology
REVIEW RETURNED	28-Jun-2021

GENERAL COMMENTS	Minor comments:  I would like to know if your project was approved by a Bioethic and Scientific Committee, if it is the case, please add the information and also if patients signed informed consent. To include data as use of drugs, alcohol, or cigarette among TB patients (In the table 1 or as extra Table). This could have a potential impact in your analysis. In the section "Survey tools and validation", you described that the 24 hours recall tools was used to obtain the patients' information. Peter S et al (Nutrients 2015, 7, 10491-10500.
--

	https://doi.org/10.3390/nu7125547) reported that this tool has limitations, such as:  • One record is seldom representative of a person's usual intake • Under-reporting/over-reporting occurs • Relies on memory • Omissions of dressings, sauces, and beverages can lead to a low estimate of energy intake Thus, how your team avoids these limitations? 4) Due to multi-drug-resistant TB requires an extended time of treatment compared with drug-sensible TB. Do have an impact the treatment scheme and time on the weight gain when are comparing these TB groups? 5) In table 2, why the Kcal value from rice is always the same? Similar question for HIV co-infected patients.
--	--

REVIEWER	Leah Jarsberg University of California San Francisco, Division of Pulmonary and Critical Care Medicine and UCSF Center for Tuberculosis
REVIEW RETURNED	14-Jul-2021

GENERAL COMMENTS	The study examines nutritional intake of 28 inpatients at a TB sanatorium along the Myanmar-Thailand border. Participants were selected in a cross-sectional design intended to proportionally represent the local TB population. Outcomes assessed were: total nutritional intake and change in BMI between diagnosis and survey, in relation to recommended intake and BMI. Despite the inpatient setting, most patients did not meet recommended guidelines. Despite its small size, this is a valuable addition to TB & nutrition science. It involves detailed work on calculating nutritional intake, and is successful as a pilot or proof-of-concept that should lead to further work on this topic. The methodology is unclear.  1. Line 81 references a 3-6 month length of TB treatment. Line 103 references routine [TB] treatment. I am not aware of any routine 3-month TB regimens; please specify what regimens were given. 2. The survey was not administered to every participant at the same week of treatment, so some patients may have had more time to improve (e.g. gain weight). Please specify the range (minimum and maximum) of "duration of treatment" values, to reassure the reader that at least all surveys were administered during the continuation phase of treatment. 3. Lines 102-103 and 246 reference inclusion criteria: some proportion of patients with extrapulmonary disease and multi-drug resistant disease, but these characteristics are not described anywhere in the results or tables. Please be explicit how many had these types of disease. 4. There is some confusion within the presentation of micronutrients.  a. Lines 165-166 state that "six selected micronutrients" were studied. More than 6 are presented in Tables 2-3 (Vitamins A, B-complex, C; minerals Ca, Fe, Na, Zinc). b. Please spell out the mineral names in the tables. c. It is my understanding that folate is part of the Vitamin B complex; in tables, please group folate with the vitamins. 5. Summary statistics methods are unclear  a. Line 162 states that "data were assessed for normality...and summarized..." Usually the normality of residuals is assessed in a regression context. Clearer wording might be, "Distributions of the data were assessed and summarized..." b. Lines 162-163 state that "data were summarized using mean and
--

	standard deviation or median and range...” Table 1 and lines 187-190 present means with 95%CI. Please either present standard deviations, or state in the methods how you calculated the 95%CI. Table 2 title states that median is presented with IQR. Please revise the methods to correspond. Additions to text or tables Please add a statement to say if the dietary support/rations are culturally similar to what patients typically prefer to eat, and specify if patients are given pre-defined portions based on nutritional guidelines. In limitations, please add something about the small sample size; line 246 references the broad inclusion criteria, but this would not be a limitation if the number of patients were larger, or if they had been sampled multiple times over the course of treatment. Table 1 should be structured similarly to Tables 2-3, with male and female columns to the right, rather than in rows below the totals.
--	---

VERSION 1 – AUTHOR RESPONSE

Reviewer: 1

Dr. Zekariyas Sahile, Ambo University

Comments to the Author:

The researchers have selected a good and interesting research title. This type of research is highly needed for the scientific community. However, this research manuscript is not clear methodologically and may not be replicable. The below list of comments may help the researchers to revise the manuscript.

Response: Thank you for your comprehensive review and very good feedback and comments.

Abstract

1. The study design was not reported in the abstract section.

Response: Thank you—this wasn’t specifically pointed out as a separate section in the abstract per BMJ Open guidelines, but we have placed this in the “Participants” section.

2. The researchers need to check the English grammar in the abstract section.

Response: The grammar has been double-checked by native English speaking team members. Please let us know specifically if there are issues.

3. The researcher did not report how the participants were selected, was it purposive sampling.

Response: This is now reported in the abstract and methods (lines 33 and 125).

4. A total of 28 participants were enrolled in the study and 20 were male and 5 were HIV coinfecting. However, the researcher was mentioned there were 118 participants under TB treatment during this study is conducted. What was the scientific base for small size?

Response: This was related to the ability to conduct the study in a timely fashion. As this was not a controlled study, there was no formal sample size calculation. This summarizes nutritional intake on a subset of TB patients receiving care at the sanatorium.

5. The primary and secondary outcome measure: the reader may not be clear on which outcome is primary and which one is secondary in the Abstract section.

Response: This has been updated (lines 36-38).

6. The authors reported they were considered sex, age, and TB/HIV co-infection in participant selection from historical records. What other factors like relapsed TB, MDR-TB, and other non-communicable diseases such as diabetes mellitus which is common TB patients. One advantage of stratification is to get an adequate sample from each stratum. However, the researcher report there was a significant difference in the proportion of male and female.

Response: To better control for factors such as these, relapsed TB patients and those with comorbidities such as diabetes mellitus were not included, as their treatment and nutritional recommendations may differ from what is commonly prescribed for TB patients. The significant difference in gender in our study is representative of the greater number of TB infections among men in the general population.

7. The researchers report there was no significant difference in macro or micronutrient intake by sex or for patients supplementing meal rations. only 4 participants were female. Do you think it appropriate to do a statistical analysis? Did the researchers check the statistical assumptions? Was measure of association the aim for this study? The researcher also said a statistically significant higher proportion of women (8/8) met caloric requirements versus men (9/20) and report P-value. Which statistical analysis the researcher performed for this report?

Response: We performed statistical analysis and reported p-values for completion, according to the BMJ Open guidelines. We understand that the sample sizes are small and that statistical significance may not always help interpretation. Chi-square test was performed for this particular comparison. The reviewer may refer to the edits we have now made to the "Statistical Analysis" section of methods for more details on statistics used in this study.

8. In the abstract section, this statement is not clear "From diagnosis to the time of 43 the survey, although participants had significant weight gain averaging 6.48 kg (95% CI: 44 3.87,

9.10) and 2.47 kg/m² (95% CI: 1.45, 3.49) in BMI (p=0.0001 for both), 50% (14/28) 45 still had mild or more severe forms of malnutrition.”

Response: This has been edited for clarity (lines 45-46).

9. The researcher's conclusion is very general in the abstract section. It would be good to focus on which macro or micronutrients were not met the nutritional requirement?

Response: The specifics have been presented in the “Results” section of the manuscript. Since participants had trouble meeting most nutritional requirements, for word count limits in the abstract, we have left this a bit general.

10. It is good to include a very clear description of nutritional support provided for the patients. Because half of the participants had not to access additional sources. I would suggest if the researcher's focus is on the nutritional support and met the nutritional requirement

Response: This information has now been included in a subsection of the Methods, “Nutrition support provided” (lines 135-150).

Strength and limitation

11. The researcher reports they used a strong methodology and accurate assessment of nutritional intake as a strength of the study. In meantime, they report they have not used the more precise measure of nutritional intake as a limitation.

Response: We have tried to clarify this, but what is meant is that we have used a robust methodology according to our means, but there may be more precise measurements of nutritional intake that one can consider in a study design such as ours. Edits to the “Strengths and limitations” section are found in lines 72-76.

Method section

12. BMI classification in the method section needs a reference.

Response: This has been added (line 190).

13. The recommended dietary intake for infected Tuberculosis patients should not be the same as healthy individuals. Did the researcher use the same RDA as healthy individuals? This is not described in the report. The researcher calculated the protein for macronutrient requirement for TB and TB/HIV by sex. The recommended requirement for carbohydrates and fat were not included in the manuscript report.

Response: Please refer to the second paragraph of “Anthropometric measurements and determining nutritional requirements” (lines 199-208). Recommended nutritional intake was calculated for each individual participant as outlined. Macronutrients (carbohydrates, fats, proteins) are adjustments typically made for TB patients’ requirements as described and referenced.

14. What is the scientific base for selecting 6 micronutrients?

Response: This was a limitation of the free nutritional analysis software that we were able to use.

15. Which specific statistical analysis used to find associations? It is not clearly described in the statistical analysis section.

Response: We have edited the “Statistical Analysis” section of the Methods for greater clarity (lines 209-222).

Result

1. The researcher reported that all participants were received two full meals per day and refreshments (processed foods and liquids). Half the participants ate the ration only from the clinic. If this information is known what is the need to ask the patient for a 24-hour recall?

Response: In addition to the type of food, we needed to measure the amount to get a full account of nutritional intake.

2. I think it would be good if the researchers were assessed the nutritional supplementation was met the nutritional requirement for Tuberculosis patients and its impact. Because, as I understand, most of the patients in that setting are mostly dependent on nutritional support provided by the clinic.

Response: This actually is the main thrust of the study—to understand where TB patients may be deficient in the diet provided and how it may be improved.

Reviewer: 2

Prof. Leslie Chavez-Galan, Instituto Nacional de Enfermedades Respiratorias
Comments to the Author:

Minor comments:

1. I would like to know if your project was approved by a Bioethic and Scientific Committee, if it is the case, please add the information and also if patients signed informed consent.

Response: Thank you, for this comment. The project was indeed reviewed and approved by the Research Ethics Committee of the Faculty of Medicine, Chiang Mai University. This additional detail has been added to the original manuscript under the subheading “Ethics approval” (line 344).

2. To include data as use of drugs, alcohol, or cigarette among TB patients (In the table 1 or as extra Table). This could have a potential impact in your analysis.

Response: Yes, this was surveyed but the participants did not self-report that they had taken any of these substances. This is now reported in lines 248-249.

3. In the section "Survey tools and validation", you described that the 24 hours recall tool was used to obtain the patients' information. Peter S et al (Nutrients 2015, 7, 10491-10500. <https://doi.org/10.3390/nu7125547>) reported that this tool has limitations, such as:
 - One record is seldom representative of a person's usual intake
 - Under-reporting/over-reporting occurs
 - Relies on memory
 - Omissions of dressings, sauces, and beverages can lead to a low estimate of energy intake. Thus, how your team avoids these limitations?

Response: These are important points. Please see that we have addressed many of these concerns in “Survey tools and validation” of the Methods section. Recall bias, under-reporting and/or over-reporting were addressed according to the literature regarding “portion size estimations.” In addition, the second paragraph of “Calculating Nutritional Intake” limited potential omissions of dressings/sauces/beverages, as the recipes of the main meals eaten by the participants were well known. Thank you for this reference which we now mention as a limitation in our Discussion section (lines 325-327).

4. Due to multi-drug-resistant TB requires an extended time of treatment compared with drug-sensible TB. Do you have an impact on the treatment scheme and time on the weight gain when comparing these TB groups?

Response: Please note that the surveys were done at one time point while participants were undergoing treatment. Table 1 summarizes their duration of treatment, which demonstrates some variability in the participants' time receiving treatment. However, upon further analysis, the time spent in treatment showed no significant association with nutritional intake.

5. In table 2, why the Kcal value from rice is always the same? Similar question for HIV co-infected patients.

Response: Thank you for pointing this out. We have double-checked these numbers and they are indeed correct. Although the means differ, they are still not significantly so. We

have reported median due to the distribution of the data. This may relate to the small sample size, the method of measurement for rice intake, or that all patients had roughly similar kcal from rice.

Reviewer: 3

Dr. Leah Jarsberg, University of California San Francisco

Comments to the Author:

The study examines nutritional intake of 28 inpatients at a TB sanatorium along the Myanmar-Thailand border. Participants were selected in a cross-sectional design intended to proportionally represent the local TB population. Outcomes assessed were: total nutritional intake and change in BMI between diagnosis and survey, in relation to recommended intake and BMI. Despite the inpatient setting, most patients did not meet recommended guidelines. Despite its small size, this is a valuable addition to TB & nutrition science. It involves detailed work on calculating nutritional intake, and is successful as a pilot or proof-of-concept that should lead to further work on this topic.

Thank you for this kind review and constructive feedback.

- 1) The methodology is unclear.
 - a) Line 81 references a 3-6 month length of TB treatment. Line 103 references routine [TB] treatment. I am not aware of any routine 3-month TB regimens; please specify what regimens were given.

Response: Thank you for catching this, this was an error on our part. The TB regimen provided is standardized—RIPE therapy (rifampicin, isoniazid, pyrazinamide, ethambutol) with initial phase of all drugs (2 months) followed by a continuation phase of isoniazid and rifampicin only (≥ 4 months). This was not the main thrust of the paper and we have included patients with HIV co-infections or extra-pulmonary TB which alters their treatment plan. Hence, in the interest of space, we have summarized this to a line in the methods, “Routine treatment was provided to all patients according to WHO guidelines” (lines 131-132).

- 2) The survey was not administered to every participant at the same week of treatment, so some patients may have had more time to improve (e.g. gain weight). Please specify the range (minimum and maximum) of “duration of treatment” values, to reassure the reader that at least all surveys were administered during the continuation phase of treatment.

Response: Thank you for this comment. We have added to the text and changed the variable name to “Treatment length at time of survey” for better clarity. We have updated the statistical analysis section and report the results as called for by the journal (mean and 95% CI). As outlined in the Statistical Analysis section, we tested the association between treatment length at time of survey and primary and secondary outcome measures and found no association.

- 3) Lines 102-103 and 246 reference inclusion criteria: some proportion of patients with extrapulmonary disease and multi-drug resistant disease, but these characteristics are not described anywhere in the results or tables. Please be explicit how many had these types of disease.

Response: Thank you for pointing this out. We have added a line to the results (lines 250-251) and the footnotes to Table 1. Although these conditions were not excluded in sampling, no participants had MDRTB and only one had extrapulmonary TB.

- 4) There is some confusion within the presentation of micronutrients.
- i) Lines 165-166 state that “six selected micronutrients” were studied. More than 6 are presented in Tables 2-3 (Vitamins A, B-complex, C; minerals Ca, Fe, Na, Zinc).

Response: Thank you—we have edited the text for consistency.

- ii) Please spell out the mineral names in the tables.

Response: As suggested by the Reviewer, all mineral names have been spelled out in the tables.

- iii) It is my understanding that folate is part of the Vitamin B complex; in tables, please group folate with the vitamins.

Response: Thank you for these points, we have updated the tables.

- 5) Summary statistics methods are unclear
- i) Line 162 states that “data were assessed for normality...and summarized...” Usually the normality of residuals is assessed in a regression context. Clearer wording might be, “Distributions of the data were assessed and summarized...”
 - ii) Lines 162-163 state that “data were summarized using mean and standard deviation or median and range...” Table 1 and lines 187-190 present means with 95%CI. Please either present standard deviations, or state in the methods how you calculated the 95%CI. Table 2 title states that median is presented with IQR. Please revise the methods to correspond.

Response: The “Statistical analysis” section of the text and Table 1 has been updated according to the reviewers’ comments. 95% CI for normally distributed, continuous data, is what is requested by BMJ Open guidelines.

Additions to text or tables

- 6) Please add a statement to say if the dietary support/rations are culturally similar to what patients typically prefer to eat, and specify if patients are given pre-defined portions based on nutritional guidelines.

Response: Thank you, based on this and another comment from a reviewer, we have added a small paragraph outlining the particulars of the rations provided to the participants in the Methods section, "Nutrition support provided" (lines 135-150).

- 7) In limitations, please add something about the small sample size; line 246 references the broad inclusion criteria, but this would not be a limitation if the number of patients were larger, or if they had been sampled multiple times over the course of treatment.

Response: We have placed this as one of the main limitations under "Strengths and limitations" (line 74).

- 8) Table 1 should be structured similarly to Tables 2-3, with male and female columns to the right, rather than in rows below the totals.

Response: Thank you, Table 1 has now been updated as well as to your above comments.

VERSION 2 – REVIEW

REVIEWER	Leslie Chavez-Galan, Instituto Nacional de Enfermedades Respiratorias, Integrative Immunology
REVIEW RETURNED	13-Oct-2021

GENERAL COMMENTS	Manuscript has been improved.
-------------------------------

REVIEWER	Leah Jarsberg University of California San Francisco, Division of Pulmonary and Critical Care Medicine and UCSF Center for Tuberculosis
REVIEW RETURNED	12-Nov-2021

GENERAL COMMENTS	Though the population description notes that patients who were unable to consent were excluded, there should be an explicit statement that patients did consent to be in the study, or that the requirement for individual consent was waived by the TCAB. Line 241 recommends that proteins should be "animal-based if possible," but animal vs plant protein was not part of this study, and there is no nutritional reference cited. Please drop this phrase if it cannot be scientifically supported. A few typos should be corrected. Please review numbers for accuracy. Line 185 "actual" Line 199 "p=630"
--

VERSION 2 – AUTHOR RESPONSE

Reviewer: 2

Prof. Leslie Chavez-Galan, Instituto Nacional de Enfermedades Respiratorias

Comments to the Author:

Manuscript has been improved.

Reviewer: 3

Dr. Leah Jarsberg, University of California San Francisco

Comments to the Author:

Though the population description notes that patients who were unable to consent were excluded, there should be an explicit statement that patients did consent to be in the study, or that the requirement for individual consent was waived by the TCAB.

We have updated this in line 119-141. This is also referred to as part of the back matter in lines 412-416.

Line 241 recommends that proteins should be "animal-based if possible," but animal vs plant protein was not part of this study, and there is no nutritional reference cited. Please drop this phrase if it cannot be scientifically supported.

Thank you, we have deleted this clause.

A few typos should be corrected. Please review numbers for accuracy.

Line 185 "actual"

Line 199 " $p=630$ "

Please note that the first typo has been edited out due to the editorial team's suggestion for this paragraph. The second typo has been corrected.